# In situ redox reactions facilitate the assembly of a mixed-valence metal-organic nanocapsule

Asanka S. Rathnayake [1], Hector W.L. Fraser[2], Euan K. Brechin[2], Scott J. Dalgarno[3], Jakob E. Baumeister[1], Joshua White[1], Pokpong Rungthanaphatsophon[1], Justin R. Walensky[1], Charles L. Barnes[1], Simon J. Teat [4] & Jerry L. Atwood [1]

C-alkylpyrogallol[4]arenes (PgCs) have been studied for their ability to form metal-organic nanocapsules (MONCs) through coordination to appropriate metal ions. Here we present the synthesis and characterization of an $Mn^{II}/Mn^{III}$-seamed MONC in addition to its electrochemical and magnetic behavior. This MONC assembles from 24 manganese ions and 6 PgCs, while an additional metal ion is located on the capsule interior, anchored through the introduction of bridging nitrite ions. The latter originate from an in situ redox reaction that occurs during the self-assembly process, thus representing a new route to otherwise unobtainable nanocapsules.

[1] Department of Chemistry, University of Missouri, 601, S. College Ave., Columbia, MO 65211, USA. [2] EaStCHEM School of Chemistry, The University of Edinburgh, David Brewster Road, EH9 3FJ Edinburgh, UK. [3] Institute of Chemical Sciences, Heriot-Watt University, Riccarton, Edinburgh EH14 4AS, UK. [4] Advanced Light Source, Lawrence Berkeley National Laboratory, 1, Cyclotron road, MS6R2100, Berkeley, CA 94720, USA. Correspondence and requests for materials should be addressed to J.L.A. (email: atwoodj@missouri.edu)

Polynuclear clusters of transition metal ions play a vital role in biologically active sites[1–6]. Over the past few decades, there has been great effort invested in the synthesis of molecular analogs of such species, but also developing an understanding of their physiological role in nature[7–12]. These assemblies have also been used to investigate structure-property relationships[10,13–19], and in this regard, clusters containing paramagnetic metal ions have attracted intense interest due to their fascinating magnetic properties[20–25]. For example, manganese-based polynuclear assemblies have been studied intensively due to their relevance in multi-disciplinary areas ranging from bioinorganic chemistry through to molecular magnetism[14,21,26–32].

Recent work in our group has focused on the synthesis of metal-organic assemblies, which arise from the utilization of *C*-alkylpyrogallol[4]arenes (PgC$_n$, where *n* represents pendant alkyl chain lengths) as ligands/structural platforms. PgC$_n$s are cyclic polyphenols which, upon deprotonation, may act as polydentate chelates for a variety of metal ions. Combination of these components under different reaction conditions gives rise to self-assembled capsules in which the stoichiometries of metal ion(s) and PgC$_n$ vary markedly[33–35]. Within this library of structures we are particularly interested in spherical metal-organic nanocapsules (MONCs). In previously published examples, the PgC$_n$ upper-rim phenolic groups are deprotonated, and either 2 or 6 PgC$_n$s coordinate to 8 or 24 divalent metal ions, respectively; these capsule types are herein referred to as dimeric or hexameric MONCs based on the number of PgC$_n$s incorporated[34–37]. MONCs containing metal ions in different oxidation states have also been reported[33,38], but these resulted from reactions involving pre-existing differences in oxidation states prior to assembly[38].

Here we report our recent findings in which we were able to synthesize a remarkable mixed-valence manganese-seamed MONC of formula [Mn$^{II}_{21}$Mn$^{III}_4$(PgC$_5$)$_6$(μ$_2$-NO$_2$)$_3$(μ$_3$-NO$_2$)$_3$(H$_2$O)$_{36}$], **1** (Fig. 1). This assembly is achieved via an in situ redox reaction, the result of which is anchoring of an additional metal ion on the capsule interior by bridging nitrite ions.

## Results

**Single-crystal X-ray studies.** Nanocapsule **1** was crystallized in a trigonal cell and the structure solution was performed in the space group *R3c*. The asymmetric unit was found to contain one third of the title formula and symmetry expansion afforded the view of the hexameric capsule shown in Fig. 1. Inspection of the structure reveals that, as is the case for other transition metal-based MONCs[34,39,40], 24 manganese ions stitch up the capsule seam, forming eight [Mn$_3$O$_3$] triangular facets. The additional manganese ion is anchored to one [Mn$_3$O$_3$] facet, linking through to three others via μ-NO$_2^-$ ions (Supplementary Figure 1a). Interestingly, this is the first MONC that has been formed with an odd number of metal ions in the assembly framework, and the fact that the system does not display extreme crystallographic disorder is notable.

All manganese ions in **1** are six-coordinate and are in one of two distinct coordination geometries. In addition, the manganese ions are in either the 2+ or 3+ oxidation state, as described in the caption to Figure 2. Oxidation states of the manganese ions in the MONC seam were assigned on the basis of bond valence sum (BVS) analysis[41] (Supplementary Table 1), and consideration of Mn——O bond lengths. In contrast, the oxidation state of the manganese ion in the interior of the capsule was deduced by taking into account both the symmetry and overall charge balance of the system. The Mn$^{II}$ ions in four of the eight facets of the assembly seam (Mn1–Mn4 and symmetry equivalents, s.e.) are coordinated to four equatorial phenolic groups and two axial water ligands, the result of which is distorted octahedral geometry (e.g., Mn1, Fig. 2a). The manganese ions in three other [Mn$_3$O$_3$] facets also exhibit distorted octahedral geometry, but with markedly different coordination spheres. In these cases, two Mn$^{II}$ ions (Mn5, Mn8, and s.e.) possess an axial water ligand on the capsule exterior, while a μ-nitrite connects the two ions on the interior (Fig. 2b). The remaining ions (Mn6 and s.e.) are in the 3+ oxidation state, and also possess an axial water molecule bonded on the capsule exterior. Further, these Mn6 ions are coordinated on the interior to another Mn$^{III}$ ion (Mn9) through a μ$_3$-nitrite ion (Fig. 2b). This nitrite ion also bridges to Mn$^{II}$ ion in the remaining [Mn$_3$O$_3$] facet (Mn7 and s.e.). The result of this is that, these Mn7 ions possess distorted trigonal prismatic geometry (Fig. 2c). The crystallographically unique Mn6 ion in the capsule seam shows clear Jahn–Teller elongation along the O12S——Mn6——O13S vector (Mn——O axial bond lengths of 2.226(7) and 2.384(6) Å, respectively) and deviation from linearity with an angle of 170.5(3)°. The Mn9 ion anchored to the Mn7/Mn7 s.e. facet is connected to three additional facets in the structure, all of which is facilitated by the μ$_3$-nitrite anions (Supplementary Figure 1b). The coordination environment around Mn9 results in a concave distortion in the capsule framework as shown in Fig. 3a. Despite this feature, **1** possesses clear $C_3$ symmetry between the three Mn$^{III}$ ions in the capsule seam, coinciding with the Mn9 position (Fig. 3b). The solid-state internal volume of **1** was calculated twice using MSRoll with probe radius of 1.25 Å: once including Mn9 with μ-nitrite ions around it (~1190 Å³), and, again, excluding all the internal components (~1360 Å³). Comparison of these volumes to that of the Cu$^{II}$-seamed nanocapsule (~1250 Å³)[34] confirms that, despite having a crowded interior environment, the void space of **1** is still quite large with respect to molecular guests.

**Electrochemistry.** As manganese(II) nitrate was used in the synthesis of **1**, it is clear that an oxidative process has taken place

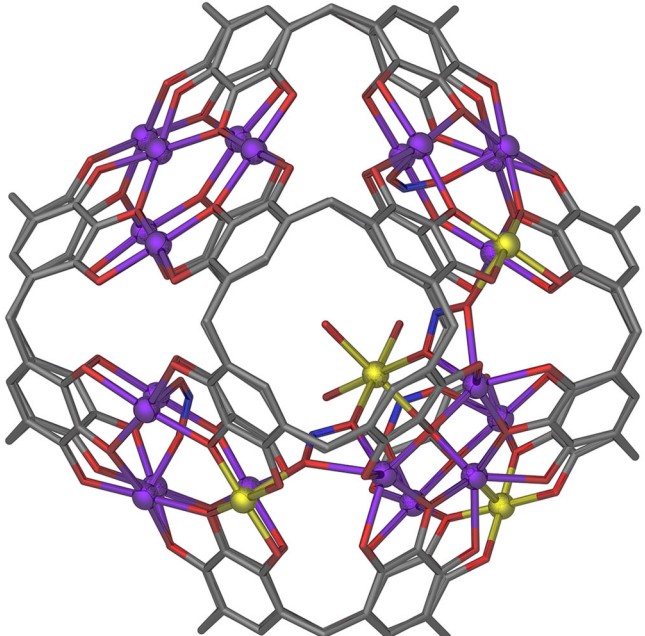

**Fig. 1** Symmetry expanded single-crystal X-ray structure of **1**. A view of **1** showing the seam of manganese ions, as well as the additional encapsulated manganese ion. Disordered PgC$_5$ alkyl chains, some axial ligands, and H atoms have been removed for clarity. Color code: Mn$^{II}$—purple, Mn$^{III}$—yellow, N—blue, O—red, and C—gray

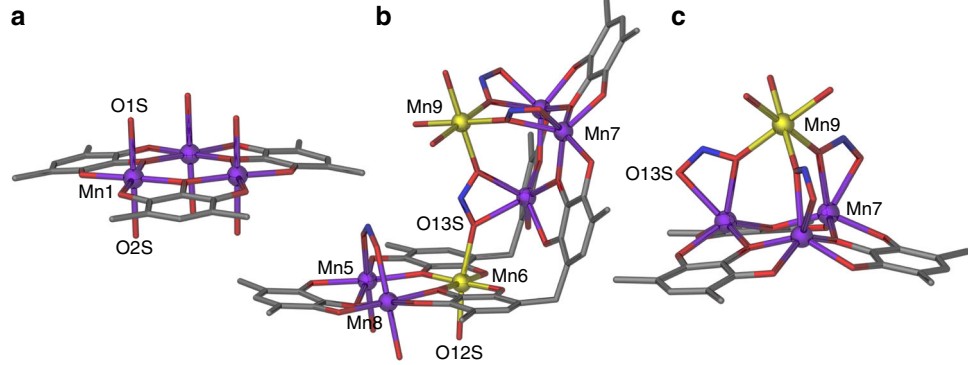

**Fig. 2** Sections of **1** showing the disparate Mn coordination sites. **a** Distorted octahedral geometry around $Mn^{II}$ ions. **b** Linkage of $Mn^{III}$ ions in the structure through $\mu_3$-nitrite anions. **c** Anchoring of Mn9 to Mn7/Mn7 s.e. facet. Disordered $PgC_5$ alkyl chains and H atoms have been removed for clarity. Color code: $Mn^{II}$—purple, $Mn^{III}$—yellow, N—blue, O—red, and C—gray

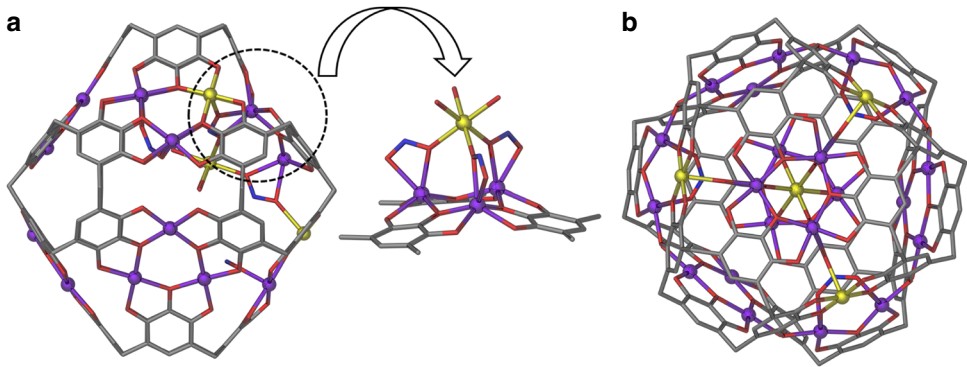

**Fig. 3** Solid-state structural features of **1**. **a** A side view of **1** showing the concave distortion on the framework. The expanded view shows the coordination of $\mu$-$NO_2^-$ ions to Mn9 that results the distortion. **b** View of **1** along the $C_3$-axis. Mn9 is the central yellow sphere. Disordered $PgC_5$ alkyl chains, some axial ligands, and H atoms have been removed for clarity. Color code: $Mn^{II}$—purple, $Mn^{III}$—yellow, N—blue, O—red, and C—gray

during self-assembly, yielding a mixture of $Mn^{II}/Mn^{III}$ ions in the MONC. The coordination of nitrite ions on the framework also demands the occurrence of a reductive process during the formation of **1**. Nitrate mediated oxidation of $Fe^{II}$ ions has previously been observed in an oxygen depleted environment[42]. Similarly, under reaction conditions described herein, nitrate ions may have acted as electron acceptors for the oxidation of $Mn^{II}$ ions (equation 1). The proposed electron-transfer reaction also indicates that acidic conditions resulting during synthesis may have supported this simultaneous redox process (final pH = 3.84)[43]. As such, this is the first reported example of a redox-mediated synthesis/assembly of a $PgC_n$-MONC.

$$NO_3^- + 2Mn^{II} + 2H_3O^+ \rightarrow NO_2^- + 2Mn^{III} + 3H_2O. \quad (1)$$

Given the above, we studied the electrochemical properties of **1** via cyclic voltammetry to support our hypothesis. Complex **1** displays a single cathodic peak, $E_{pc}$, in acetonitrile at −0.83 V consistent with the reduction of $Mn^{III}$ (Fig. 4a). The peak appears to be irreversible due to the lack of a complementary anodic peak following reduction. A single anodic peak is observed in the cyclic voltammogram of **1** in propylene carbonate with an $E_{pa}$ value of +0.64 V (Fig. 4b). The anodic peak is assigned to the oxidation of $NO_2^-$ to $NO_2$[44]. This is comparable to the results obtained in other solvents including protic media, DMSO, and ionic liquids[45–49]. No concomitant reduction peak is observed, leading to the conclusion that this process is irreversible. Similar

electrochemical behavior was observed for **1** in $CH_2Cl_2$: a single anodic peak with an $E_{pa}$ of +0.87 V with no corresponding cathodic peak (Fig. 4c). Overall, this electrochemical behavior supports that the $NO_3^-$ ions have acted as electron acceptors during the formation of **1**.

**Magnetism.** DC magnetic susceptibility measurements were carried out on a powdered polycrystalline sample of compound **1** in an applied magnetic field of $H = 0.1$ T over the temperature range $T = 2$–300 K. Figure 5 shows the experimental data plotted as the $\chi_M T$ product vs. $T$, where $\chi_M$ is the molar magnetic susceptibility. The value of $\chi_M T$ at $T = 300$ K is ~91 $cm^3$ K $mol^{-1}$, lower than that expected for the sum of the Curie constants for 21 non-interacting $Mn^{II}$ ($s = 5/2$) ions and 4 non-interacting $Mn^{III}$ ($s = 2$) ions, with $g = 2.00$ (103.875 $cm^3$ K $mol^{-1}$). As the temperature decreases, the magnitude of $\chi_M T$ decreases steadily, reaching a value of 20.96 $cm^3$ K $mol^{-1}$ at $T = 2$ K. This behavior is indicative of the presence of (relatively weak) antiferromagnetic (AF) exchange interactions between the constituent metal ions, as would be expected for a large cluster containing multiple $Mn^{II}$ and $Mn^{III}$ ions[50]. The presence of the symmetry breaking, $\mu$-$NO_2^-$ anions, and the large nuclearity of the cage prevents any detailed quantitative analysis of the data. However, a fit of the linear section of the $1/\chi_M$ vs. $T$ data (Fig. 5 inset; 300–30 K) affords Curie and Weiss constants of $C = 102.04$ $cm^3$ K $mol^{-1}$ and $\theta = -38.3$ K, respectively. Magnetization data collected for **1** in the $H = 0$–7 T field range at temperatures between $T = 2$–7 K

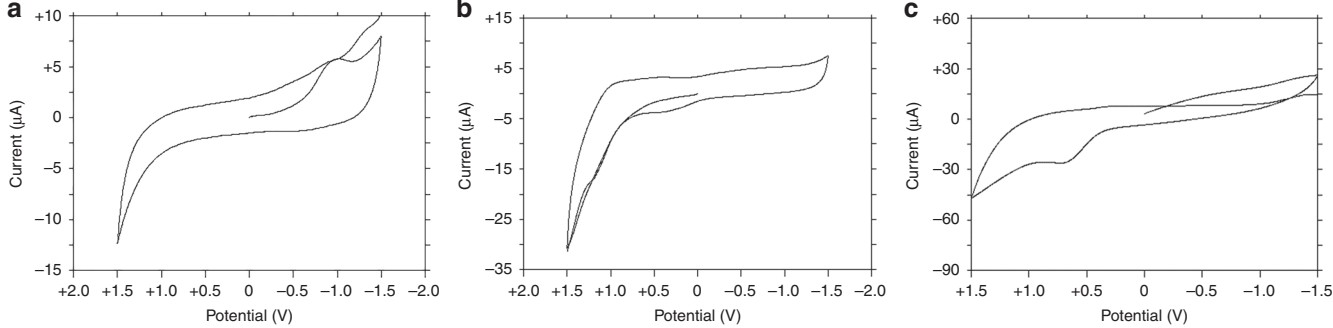

**Fig. 4** Electrochemical behavior of **1**. Cyclic voltammograms of **1** in (**a**) acetonitrile, (**b**) propylene carbonate, and (**c**) dichloromethane with 0.1 M TEAP. Scan rate is 100 mV s$^{-1}$

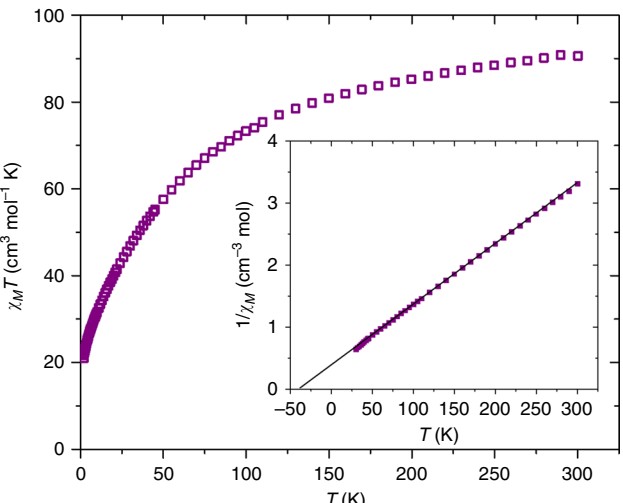

**Fig. 5** Magnetic susceptibility of **1** as a function of temperature. The magnetic susceptibility data have been measured in an applied magnetic field of $H = 0.1$ T over the temperature range $T = 2$-300 K. The data is plotted as $\chi_M T$ vs. $T$ and shows characteristic antiferromagnetic behavior. The inset shows the $1/\chi_M$ vs. $T$ data; the solid line is a fit of the experimental data to the Curie–Weiss law.

are consistent with the presence of weak AF exchange, the value of $M$ increasing in a near linear-like fashion with increasing $H$, without reaching saturation (Supplementary Figure 2).

## Discussion

To conclude, we have presented the synthesis and structural characterization of a mixed-valence Mn$^{II}$/Mn$^{III}$-seamed MONC. Electrochemical and magnetic behaviors of this remarkable assembly have been studied and, unlike other previously reported PgC$_n$-MONCs, **1** contains an odd number of metal ions with the additional ion residing on the capsule interior. The degree of crystallographic order in the system is noteworthy. We propose that bridging NO$_2^-$ ions arise from a NO$_3^-$ mediated oxidation of Mn$^{II}$ ions. Overall this is an important development since it suggests that in situ redox reactions may be used to access a vast library of new MONCs. This may be achieved through the controlled addition of different oxidizing or reducing species, all of which may influence reactions differently and provide access to new MONCs; these would arise from the generation/inclusion of transition metal centers in oxidation states that are atypical for

such species formed under more typical/ambient reaction conditions. It may also allow the fine-tuning of the coordination chemistry of the internal wall, as well as the resulting magnetic properties. The resulting MONCs may then be used in the rational construction of framework materials with controlled magnetic and/or host:guest properties. This will be subject of future work in the area, the results of which will be reported in due course.

## Methods

**Synthesis of 1**. PgC$_5$ (0.166 g, 0.2 mmol) and sodium methoxide (0.032 g, 0.6 mmol) were mixed in a 1:1 (v/v) CH$_2$Cl$_2$/ethanol mixture (10 ml each). The solution was sonicated at 45 °C for 20 min. The resulted pinkish-white turbid solution was cooled down for 10–15 min and Mn(NO$_3$)$_2$. 4 H$_2$O (0.2002 g, 0.8 mmol) was added. The solution was sonicated for an additional 30 min at 45 °C (final pH = 3.84). The resulting dark solution was set aside and black crystals formed over a period of 2 days upon slow evaporation of the mother liquor. Yield: 0.86 g = 6% (with respect to Mn).

**Single-crystal X-ray analysis**. Single-crystal data of **1** was collected on Bruker D8 with PHOTON 100 detector with synchrotron radiation. The structure was solved and refined using SHELX programs and X-SEED. In order to handle the disorder associated with solvent molecules and to model the structure appropriately, SQUEEZE program was applied to the data set.

**Electrochemistry**. Electrochemical data were obtained with a Bioanalytical Systems Inc. (BAS) CV-50 instrument. Tetraethylammonium perchlorate (TEAP; 0.1 M) was used as the supporting electrolyte for all solvents systems used. All measurements were taken using a platinum wire auxiliary electrode (BAS), a non-aqueous Ag/AgCl reference electrode (BAS), and a glassy carbon working electrode (BAS). Results were standardized against ferrocene (Fc). All experiments were carried out under argon to avoid possible air oxidation.

**Magnetism**. DC magnetic susceptibility and magnetization measurements on powdered microcrystalline sample of **1** were performed using a MPMS XL SQUID magnetometer working in the $T = 2$–300 K and $H = 0$–7 T temperature and field ranges, respectively. Diamagnetic corrections were applied to the data using Pascal's constants. The sample was stored and prepared in a glove box prior to measurement to avoid possible air oxidation.

**Data availability**. Further experimental and characterization details can be found in Supplementary Information. CCDC 1442429 contains the crystallographic data for nanocapsule **1**. Additional data are available from the corresponding author upon request.

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

## Acknowledgements

We thank Beverly B. DaGue for mass spectrometry analysis of this work.

## Author contributions

A.S.R. and J.R.W. synthesized the nanocapsule. H.W.L.F. and E.K.B. performed and analyzed the magnetic measurements. J.E.B. performed the electrochemistry experiment. P.P. prepared and handled the samples under air-free conditions. S.J.T. performed the X-ray crystallography experiment. C.L.B. analyzed and solved the crystal structure. A.S.R.,

E.K.B., S.J.D., J.R.W. and J.L.A. drafted and edited the manuscript. All authors discussed the results and commented on the manuscript.

## Additional information

**Competing interests:** The authors declare no competing interests.

