## [Peer Review File · Nature Communications]

Reviewers' comments:

Reviewer #1 (Remarks to the Author):

The manuscript by Atwood et al described the design, synthesis and characterization of a novel MnII/MnIII-seamed metal organic nanocapsules through coordination of C-alkylpyrogallol[4]arenes with appropriate metal ions. They show that this new kind of metal organic nanocapsule has been assembled by 24 manganese ions and 6 PgCs, interestingly an additional metal ion is located on the capsule interior through bridging nitrite ions. Single-crystal X-ray and electrochemical studies demonstrated that this new kind of self-assembled structure originates from an in-situ redox reaction in self-assembly process.

I think it is a new development for designing unobtainable nanocapsules.

I can recommend for publishing with a minor revising.

Some minor points:

1. In this work, the mixed-valence manganese-seamed MONC appears with a formula $[\text{Mn}^{\text{II}}21\text{Mn}^{\text{III}}4(\text{PgC}5)_6(\mu2\text{-NO}_2)_3(\mu3\text{-NO}_2)_3(\text{H}_2\text{O})_{36}]$. In the preparation of MONC, if adding a mixture of MnII (NO₂)₂ and MnIII (NO₂)₂ with PgC₅, can you get the similar MONC?
2. Although some researchers know the PgC₅, to let ones not in this filed know well, the author should give its synthesis or a reference in SI.
3. The author had better to give the details for single-crystal preparation and the pictures of these single-crystal in SI.

Reviewer #2 (Remarks to the Author):

This manuscript by Rathnayake et al. reports this first example of a metal-organic nanocapsule (MONC) to have an odd number of metal ions in the assembled unit. It is also of interest because the Mn ions in the MONC are of mixed-valence, i.e., both Mn(II) and Mn(III) are present. The authors deduce that this results from an in-situ redox reaction.

The manuscript is well written and the MONC is characterized by X-ray crystallography, electrochemistry, and DC magnetic susceptibility measurement. I recommend accepting the article for publication pending minor revisions noted below.

The authors note the high degree of crystallographic order within the system, but in all presentations of the complex the pentyl chains have been omitted. Are these also well-ordered? It would be surprising if they were. The authors should either show a complex with the pentyl chains in the supporting information or comment on whether they are ordered or not in the manuscript.

The authors propose a redox reaction (equation 1) to explain the production of the nitrite ions and the Mn(III) ions. They state that this indicates "that acidic conditions resulting during the synthesis may have supported this simultaneous process" (line 112-113). Is there experimental evidence for the acidic reaction medium? Sodium methoxide is used as a reagent. The authors should state the experimentally-determined pH of the reaction medium.

The authors state that the "electrochemical behavior confirms that the NO₃⁻ ions have acted as electron acceptors during the formation of 1." (lines 126-127) I believe this statement is too strong. It should be amended by replacing "confirms" with "supports" or "is consistent with".

It appears that the mixed-valence MONC was discovered serendipitously rather than by rational design. There is nothing wrong with that, as it shares company with many great breakthroughs. However, I would like the authors to describe general methods that might be employed to promote in-situ redox reactions in order to "access a vast library of new MONC's."

Reviewer #3 (Remarks to the Author):

In this manuscript, the first mixed-valence MnII/MnIII-PgCn-MONC was synthesized and characterized. This is an interesting paper. The authors do have some good chemistry; however X-ray crystal structure result was roughly prepared:

1) Checkcif complains a mismatch between cell contents calculated from `_chemical_formula_sum` and `_cell_formula_units_Z` and the atom sites present in the `_atom_site_` list and the `_symmetry_` information

Alert level G

CELLZ01_ALERT_1_G ALERT: Large difference may be due to a symmetry error - see SYMMG tests

atom Z*formula cif sites diff

C 1728.00 1728.00 0.00

H 1872.00 1872.00 0.00

Cl 6.00 0.00 6.00

Mn 150.00 150.00 0.00

N 36.00 36.00 0.00

O 720.00 720.00 0.00

and

PLAT041_ALERT_1_G: Calc. and Reported SumFormula Strings Differ Please Check

However, there is no any explanation in the text or cif.

2) Checkcif also complains

Alert level A

PLAT201_ALERT_2_A Isotropic non-H Atoms in Main Residue(s) 33 Report

and

Alert level B

PLAT341_ALERT_3_B Low Bond Precision on C-C Bonds 0.02044 Ang.

Also, they use SQUEEZE for the refinement.

The author claimed that "Result of disorder and limited data quality"

However, currently the resolution is up to 0.81 Å. Is there any reasons that the authors don't want to cut down the resolution, such as to 0.84 Å, IUCR standard, but keep the "bad data" during the structure refinement without carefully refining the disorder?

Title: *In-situ* Redox Reactions Facilitate the Assembly of a Mixed-Valence Metal-Organic Nanocapsule

Authors: Asanka S. Rathnayake, Hector W. L. Fraser, Euan K. Brechin, Scott J. Dalgarno, Jakob E. Baumeister, Joshua White, Pokpong Rungthanaphatsophon, Justin R. Walensky, Charles L. Barnes, Simon J. Teat, and Jerry L. Atwood

Reviewer #1 (Remarks to the Author):

The manuscript by Atwood et al described the design, synthesis and characterization of a novel Mn^{II}/Mn^{III}-seamed metal organic nanocapsules through coordination of C-alkylpyrogallol-[4]arenes with appropriate metal ions. They show that this new kind of metal organic nanocapsule has been assembled by 24 manganese ions and 6 PgC5, interestingly an additional metal ion is located on the capsule interior through bridging nitrite ions. Single-crystal X-ray and electrochemical studies demonstrated that this new kind of self-assembled structure originates from an in-situ redox reaction in self-assembly process.

I think it is a new development for designing unobtainable nanocapsules.

I can recommend for publishing with a minor revising.

Some minor points:

1. In this work, the mixed-valence manganese-seamed MONC appears with a formula $[\text{Mn}^{\text{II}}_{21}\text{Mn}^{\text{III}}_4(\text{PgC5})_6(\mu\text{-NO}_2)_3(\mu_3\text{-NO}_2)_3(\text{H}_2\text{O})_{36}]$. In the preparation of MONC, if adding a mixture of $\text{Mn}^{\text{II}}(\text{NO}_2)_2$ and $\text{Mn}^{\text{III}}(\text{NO}_2)_2$ with PgC5, can you get the similar MONC?
2. Although some researchers know the PgC5, to let ones not in this filed know well, the author should give its synthesis or a reference in SI.
3. The author had better to give the details for single-crystal preparation and the pictures of these single-crystal in SI.

Responses:

1. We assumed that the reviewer asked about the possibility of obtaining $[\text{Mn}^{\text{II}}_{21}\text{Mn}^{\text{III}}_4(\text{PgC5})_6(\mu_2\text{-NO}_2)_3(\mu_3\text{-NO}_2)_3(\text{H}_2\text{O})_{36}]$ by adding a mixture of $\text{Mn}^{\text{II}}(\text{NO}_2)_2$ and $\text{Mn}^{\text{III}}(\text{NO}_2)_3$ with PgC5:

Note: Metal nitrites are known to be strong oxidizing agents. Therefore, the following reactions have been performed using $\text{Mn}(\text{OAc})_2$ (as the source of Mn^{II} ions), $\text{Mn}(\text{OAc})_3$ (as the source of Mn^{III} ions), and NaNO_2 (as the source of NO_2^-).

Control experiment#1 – PgC_5 , $\text{Mn}(\text{OAc})_2$ and NaNO_2 was mixed in a 1:1 CH_2Cl_2 /ethanol mixture in the presence of sodium methoxide. A black solution was formed and this did not yield any crystals/precipitates.

Control experiment#2 – PgC_5 , $\text{Mn}(\text{OAc})_3$ and NaNO_2 was mixed in a 1:1 CH_2Cl_2 /ethanol mixture in the presence of sodium methoxide. A black solution was formed and this did not yield any crystals/precipitates.

Several experiments have been carried out as follows:
 $\text{Mn}(\text{OAc})_2$ and $\text{Mn}(\text{OAc})_3$ were mixed (in different ratios) with PgC_5 and NaNO_2 in the presence of sodium methoxide (in 1:1 CH_2Cl_2 /ethanol mixture). In all cases black precipitates were formed over a period of 3-4 days.

Based on the observations, $[\text{Mn}^{\text{II}}_{21}\text{Mn}^{\text{III}}_4(\text{PgC}_5)_6(\mu_2\text{-NO}_2)_3(\mu_3\text{-NO}_2)_3(\text{H}_2\text{O})_{36}]$ cannot be obtained by adding a mixture of MnII, MnIII, and NO_2^- to PgC_5 .

2. The synthesis of PgC_5 :
Thank you for this suggestion, a detailed experimental method for the synthesis of PgC_5 has been added to the supplementary information.
3. Details on single-crystal preparation and pictures of **1** in SI:
The detailed experimental method for the synthesis of **1** has been added to the supplementary information. In addition, several figures of nanocapsule **1** have also been added to SI (Supplementary Figure 1).

Reviewer #2 (Remarks to the Author):

This manuscript by Rathnayake et al. reports this first example of a metal-organic nanocapsule (MONC) to have an odd number of metal ions in the assembled unit. It is also of interest because the Mn ions in the MONC are of mixed-valence, i.e., both Mn(II) and Mn(III) are present. The authors deduce that this results from an in-situ redox reaction.

The manuscript is well written and the MONC is characterized by X-ray crystallography, electrochemistry, and DC magnetic susceptibility measurement. I recommend accepting the article for publication pending minor revisions noted below.

The authors note the high degree of crystallographic order within the system, but in all presentations of the complex the pentyl chains have been omitted. Are these also well-ordered? It would be surprising if they were. The authors should either show a complex with the pentyl chains in the supporting information or comment on whether they are ordered or not in the manuscript.

The authors propose a redox reaction (equation 1) to explain the production of the nitrite ions and the Mn(III) ions. They state that this indicates "that acidic conditions resulting during the synthesis may have supported this simultaneous process" (line 112-113). Is there experimental evidence for the acidic reaction medium? Sodium methoxide is used as a reagent. The authors should state the experimentally-determined pH of the reaction medium.

The authors state that the "electrochemical behavior confirms that the NO₃⁻ ions have acted as electron acceptors during the formation of **1**." (lines 126-127) I believe this statement is too strong. It should be amended by replacing "confirms" with "supports" or "is consistent with".

It appears that the mixed-valence MONC was discovered serendipitously rather than by rational design. There is nothing wrong with that, as it shares company with many great breakthroughs. However, I would like the authors to describe general methods that might be employed to promote in-situ redox reactions in order to "access a vast library of new MONC's."

Responses:

1. Disorder of pendant pentyl chains:

The pentyl chains of the PgC₅ molecules are disordered. However, the important part of the chemistry is the four-membered aryl moieties. In **1**, the aryl groups and metal ions in the main cage moiety are well-ordered, and we specified that as a "high degree of crystallographic order in the system". The alkyl chains are on the exterior of the molecule, and are used to aid solubility and/or affect crystal packing; we previously showed that different chain lengths alter the packing of MONCs, with some crystallizing particularly well. In order to indicate disorder in the pentyl chains we have changed the word "PgC₅ alkyl chains" to "disordered PgC₅ alkyl chains" in all figure captions. Also, a figure of **1** showing all the disordered alkyl chains has been added to SI (Supplementary Figure 1c) as suggested.

2. Experimental pH value:

The resulting pH value (pH = 3.84) has been added to the main text.

3. The word "confirm" has been replaced with "supports" in the main text.

4. General methods that result the formation of mixed-valence MONCs:

In general, by adding appropriate oxidizing (ex- KMnO₄) or reducing (ex- NaH) agent as a reagent, there is a possibility of obtaining mixed-valence MONCs. A brief description on this has been added to the conclusion.

Reviewer #3 (Remarks to the Author):

In this manuscript, the first mixed-valence MnII/MnIII-PgCn-MONC was synthesized and characterized. This is an interesting paper. The authors do have some good chemistry; however X-ray crystal structure result was roughly prepared:

1) Checkcif complains a mismatch between cell contents calculated from `_chemical_formula_sum` and `_cell_formula_units_Z` and the atom sites present in the `_atom_site_list` and the `_symmetry_` information

Alert level G

CELLZ01_ALERT_1_G ALERT: Large difference may be due to a symmetry error - see SYMMG tests

atom	Z*formula	cif	sites	diff
C	1728.00	1728.00	0.00	
H	1872.00	1872.00	0.00	
Cl	6.00	0.00	6.00	
Mn	150.00	150.00	0.00	
N	36.00	36.00	0.00	
O	720.00	720.00	0.00	

and

PLAT041_ALERT_1_G: Calc. and Reported SumFormula Strings Differ Please Check

However, there is no any explanation in the text or cif.

2) Checkcif also complains

Alert level A

PLAT201_ALERT_2_A Isotropic non-H Atoms in Main Residue(s) 33 Report

and

Alert level B

PLAT341_ALERT_3_B Low Bond Precision on C-C Bonds 0.02044 Ang.

Also, they use SQUEEZE for the refinement.

The author claimed that "Result of disorder and limited data quality"

However, currently the resolution is up to 0.81 Å. Is there any reasons that the authors don't want

to cut down the resolution, such as to 0.84 Å, IUCR standard, but keep the "bad data" during the structure refinement without carefully refining the disorder?

Responses:

1. Apologies, this has been resolved and was due to having ClO in the moiety and sum formulas.
2. PLAT201_ALERT_2 A Isotropic non-H Atoms in Main Residue(s) 33 Report.

A number of atoms in PgC₅ alkyl chains have been refined isotropically due to disorder. Full modelling and anisotropic refinement of these disordered alkyl chains was not feasible without the use of a large number of constraints / restraints.

3. PLAT341_ALERT_3_B Low Bond Precision on C-C Bonds 0.02044 Ang.

We propose that this is due to disorder present on the exterior of the capsule, as the main framework of the system is well-ordered.

REVIEWERS' COMMENTS:

Reviewer #1 (Remarks to the Author):

The authors addressed all questions we cared, I recomend to publish at present form.

Reviewer #2 (Remarks to the Author):

In my view the the issues raised in the peer review have been satisfactorily addressed in the revised manuscript. I recommend accepting this version for publication.

Reviewer #3 (Remarks to the Author):

The authors explained why they process the refinement. If the authors believe they have done their best in this refinement and such crystallographic result without a careful disorder refinement doesn't affect the significance of the chemistry result, it will be worth publishing.